# Field and Cage Studies Show No Effects of Exposure to Flonicamid on Honey Bees at Field-Relevant Concentrations

**DOI:** 10.3390/insects13090845

**Published:** 2022-09-16

**Authors:** William G. Meikle, Milagra Weiss

**Affiliations:** Carl Hayden Bee Research Center, USDA-ARS, Tucson, AZ 85719, USA

**Keywords:** neonicotinoid, sublethal effects, colony-level behavior, thermoregulation, bee cage study

## Abstract

**Simple Summary:**

The effects of pesticides on honey bees are of great concern to beekeepers and the general public. In particular, a class of pesticides called neonicotinoids has been found in some studies to affect bee learning and colony function. A pesticide classified as a neonicotinoid, flonicamid, has been detected in the residue analyses of honey and pollen samples for a published study, and that compound was tested at three concentrations (0, 50, and 250 parts per billion (ppb)) in sugar syrup in a field study with bee colonies and in two cage studies. Colony population levels, hive weight changes, thermoregulation, and CO_2_ concentrations were measured in the field study, and syrup consumption, thermoregulation, and survivorship by worker bees were measured in the cage studies. No significant treatment effects of the pesticide exposure were observed in either the field or the cage studies. These results support the idea that flonicamid is safe for honey bees at these concentrations. Publishing such results when they occur is important, so people use safe pesticides when they need them.

**Abstract:**

The extent to which insecticides harm non-target beneficial insects is controversial. The effects of long-term exposure on honey bees to sublethal concentrations of flonicamid, a pyridinecarboxamide compound used as a systemic insecticide against sucking insects, were examined in a field study and two cage studies. The field study involved the continuous weight, temperature, and CO_2_ monitoring of 18 honey bee colonies, 6 of which were exposed over six weeks to 50 ppb flonicamid in sugar syrup, 6 exposed to 250 ppb flonicamid in syrup, and 6 exposed to unadulterated syrup (control). Treatments were derived from concentrations observed in honey samples in a published study. No effects were observed on foraging activity, hive weight gain, thermoregulation, or average CO_2_ concentrations. However, Varroa mite infestations may have also contributed to experimental variability. The two cage studies, in which cages (200 newly-emerged bees in each) were exposed to the same flonicamid concentrations as the field study and kept in a variable-temperature incubator, likewise did not show any experiment-wide effects on survivorship, thermoregulation, or syrup consumption. These results suggest that field applications of flonicamid that result in concentrations as high as 250 ppb in honey may be largely safe for honey bees.

## 1. Introduction

Honey bees are often exposed to insecticides, yet the extent to which they are harmed by those insecticides remains controversial [1,2,3]. Neonicotinoid pesticides have been a particular object of controversy. Neonicotinoids are agonists of the nicotinic acetylcholine receptor (nAChR), causing insect paralysis and death [4]. Neonicotinoids typically have a broad spectrum of insecticidal activity, low mammalian toxicity, and versatility in application methods [5]. Some countries have banned neonicotinoid insecticides due to concerns about the impact of residues on pollinators such as honey bees [6], and those bans have, in turn, increased concerns about the potential impacts of insecticides used to replace neonicotinoids [1,7].

Flonicamid, a pyridinecarboxamide [8], has been categorized as a neonicotinoid insecticide [9], although the mode of action is different from that of other neonicotinoids [10]. Flonicamid is a systemic insecticide with selective activity against thrips and hemipterous pests, such as aphids and whiteflies [11]. Studies involving aphids showed that flonicamid caused starvation by inhibiting stylet penetration to plant tissues [11]. The LD_50_ for honey bees has been reported to be about seven parts per million (ppm) [12], although another source reported acute oral and contact LD_50_ values in excess of 100 µg per bee (about 1000 ppm) [13]. Honey bees can be exposed extensively to sublethal concentrations during foraging. In a study on the effects of landscapes on honey bee colony health conducted in southern California, flonicamid was detected in 6 of 24 honey samples at concentrations of up to 20 parts per billion (ppb) and in 6 of 33 bee bread (stored pollen) samples in concentrations up to 42 ppb [14]. Other neonicotinoid pesticides, e.g., imidacloprid and clothianidin, have been shown to have a measurable impact on colony behavior at concentrations as low as 5 ppb [15,16,17,18].

The focus of this study was the effect of exposure of flonicamid on honey bee colony behavior in a field experiment and on honey bee survivorship and thermoregulation in controlled cage studies, as has been observed with those other pesticides mentioned above. The sensor-based field methods described here have been used to detect sublethal effects of a variety of pesticides on honey bee colony growth and activity [14,15,16,19]. The study site in southern Arizona typically has low rainfall (usually <300 mm per year) with highly seasonal nectar flows. By feeding colonies adulterated syrup during a nectar dearth, the confounding effects of alternative sources of nectar can be reduced or removed, see [15,16,19]. Cage studies, using temperatures that cycle every 24 h, have previously been shown to be effective at showing treatment effects with the neonicotinoid imidacloprid [20].

## 2. Materials and Methods

### 2.1. Syrup Preparation

Control (0 ppb flonicamid) sucrose solution was mixed at 1:1 *w*:*w* (e.g., 500 g sucrose:500 mL distilled water). Sucrose was added to distilled water in a 5-gallon bucket and mixed using an electric drill with a mortar mixing attachment until sugar was completely dissolved. Sucrose solution for solutions with flonicamid (PESTANAL, CAS# 158062-67-0) was mixed in the same manner but 50 mL was withheld to allow for the added volume of respective flonicamid spikes. Five hundred grams of sugar was dissolved in 450 mL of distilled water to allow for the addition of a 50 mL spike to achieve 1 kg of treatment solution. Nine hundred fifty grams of sugar solution was transferred to a Nalgene bottle, then the 50 g was spike added to each individual bottle. A 10 ppm flonicamid stock solution was made by dissolving 1.0 mg of flonicamid, in 100 mL of distilled water, using a mixing bar but without heat. To avoid problems with static electricity, the flonicamid was weighed into small, nonreactive plastic receptacles, those receptacles were placed in the solution, the solution was stirred, and the receptacles were removed after the flonicamid had dissolved. For the 50-ppb solution: 5 mL of the stock solution was mixed into 45 mL of distilled water to achieve 50 mL of spike solution, which was then added to 950 g of the short sucrose solution to achieve 1 kg of 50-ppb flonicamid syrup. For the 250-ppb solution, 25.0 mL of stock solution was mixed into 25.0 mL of distilled water, and that solution was added to 950 g of the short solution to achieve 1 kg of 250-ppb flonicamid syrup.

### 2.2. Field Experiment

In March 2019, 18 bee colonies were obtained with marked European queens (Olivarez Honey Bees, Inc. Orland, CA 95963, USA), each containing at least 1 kg honey bees in painted, 10-frame, wooden Langstroth boxes (43.7 L capacity) with migratory wooden lids, and installed in the Santa Rita Experimental Range (31°47′2″ N, 110°51′37″ W, elevation 1200 m). Each hive had 2–4 frames with sealed brood and was given a 1-frame feeder and a second Langstroth box as a super. Hives were placed on stainless steel electronic scales (Tekfa model B-2418 and Avery Weigh-Tronix model BSAO1824-200) (max. capacity: 100 kg, precision: ±20 g; operating temperature: −30 °C to 70 °C) and linked to 16-bit dataloggers (Hobo UX120-006M External Channel datalogger, Onset Computer Corporation, Bourne, MA, USA) with weight recorded every 5 min. The system had an overall precision of approximately ±20 g. Hives were arranged in groups of 6 hives near a central box containing electronic equipment and all hives faced south to reduce the effects of hive direction on daily colony behavior. Hives within such a group were 0.5–1 m apart and groups were >3 m apart. On 1 July, a temperature sensor (Thermochron iButtons, Maxim Integrated, San Jose, CA, USA, precision ±0.06 °C) enclosed in plastic tissue embedding cassettes (Thermo Fisher Scientific, Waltham, MA, USA) was stapled to the center of the top bar on the 5th frame in the bottom box of each hive and set to record every 30 min (a reduced sampling rate was used because of the limited data storage capacity of those sensors). CO_2_ probes (model GMP251, Vaisala Inc., Helsinki, Finland), calibrated for 0–20% concentrations, were placed on top of the center frames in the top box of each hive and linked to UX120-006M dataloggers set to record every 5 min. Fewer probes were available from 13 September–23 October, so the number of monitored hives decreased at that time across all treatment groups.

Colonies were all fed 2 kg sugar syrup (1:1 *w*:*w*) and 200 g pollen patty, made at a ratio of 1:1:1 corbicular pollen (Great Lakes Bee Co., Fremont, MI, USA): Granulated sugar: Drivert sugar (Domino Foods, Yonkers, NY, USA). On 18 June, pieces of slick paperboard coated with petroleum jelly and covered with mesh screens were inserted onto the hive floor to monitor Varroa mite fall within the hive. The boards were removed 2 days later, and the number of mites counted on each board. Infestation levels of Varroa were again monitored post treatment in September. Colonies were treated for Varroa with amitraz (Apivar, Veto-Pharma, Palaiseau, France) on 28 June and with thymol (Apiguard, Vita Bee Health, Basingstoke, UK) on 8 October.

Hives were assessed on 11 July (pre-treatment), again on 6 September, 11 October, 15 November, and 14 February 2020 using a published protocol (see 13). Dates were chosen to obtain sufficient data without frequent colony disruption. Briefly, the hive was opened after the application of burlap smoke, and each frame was lifted out sequentially, gently shaken to dislodge adult bees, photographed using a 16.3-megapixel digital camera (Canon Rebel SL1, Canon USA, Inc., Melville, NY, USA), weighed on a portable scale (model EC15, OHaus Corp., Parsippany, NJ, USA), and replaced in the hive. Frame photographs were analyzed later in the laboratory (see below). During this first assessment, all hive components (i.e., lid, inner cover, boxes, bottom board, etc.) were also shaken free of bees and weighed. The total adult bee population weight was calculated by subtracting the combined weights of hive components free of bees from the total hive weight with bees recorded at midnight prior to the inspection. At the initial inspection, 3–5 g of honey was collected from each hive into 50 mL centrifuge tubes and stored at −20 °C to preserve them. Samples collected in September, prior to treatment, were pooled and subjected to a full panel analysis for residues of pesticides and fungicides, from all major classes, by the Laboratory Approval and Testing Division, Agricultural Marketing Service, USDA. Honey samples were pooled within treatment group and subjected only to neonicotinoid residue analysis.

After the initial assessment, hives were ranked with respect to adult bee mass and then assigned to treatment group, with 6 hives per group, while ensuring that the average colony bee masses per group were approximately equal and after eliminating assignments that resulted in excessive spatial clumping of the treatments. Just prior to treatment all broodless frames containing honey and/or pollen were replaced with frames of empty drawn comb collected earlier from the same apiary. Colonies were then fed 2–3 kg syrup twice per week from 16 July to 30 August. Syrup consumption per colony was recorded. At each of the post-treatment assessments, the same protocol was followed, but only the frames, hive lid, and inner cover were weighed, and those values were used to correct for moisture content changes in the wood and improve estimates of adult bee mass.

### 2.3. Cage Experiments

On 5 June 2021, at the Carl Hayden Bee Research Center in Tucson, AZ, several frames of mature brood were removed from each of four colonies with Cordovan Italian queens (C.F. Koehnen & Sons, Inc., Glenn, CA, USA) and placed in an incubator (Percival model I36VL) at 32 °C and 50% r.h. Adult bees emerging over the following 48 h were distributed among 28 Plexiglas^®^ cages (internal volume: 785 cm^3^) until cages had 200 bees (8 replicate cages per group). Each cage had plastic feeding bottles containing 30 mL sugar solution and 60 mL water, and a 4 × 4 cm square of wax foundation attached to a piece of screen hung vertically in the center. The sugar syrup was prepared as described above. Two iButtons were attached to the center of the square, one on either side, and programmed to record temperature every 5 min. All cages were placed in the incubator, on one of three shelves. Placement was random with respect to treatment group. Temperature sensors were also placed on each shelf in the incubator, in order to control for temperature gradients within the incubator. Pollen patty was prepared as described above. A 10 g sample of the patty was placed inside a rubber gasket accessed via a hole in the side of the cage. Two such samples were provided for the July experiment (for a total of 20 g), and three were provided for the August experiment (for a total of 30 g). Newly emerged bees were kept for one week at a constant 30 °C. Thereafter, the temperature in the incubator was set to vary with 12 h at 30 °C and 12 h at 15 °C. Dead bees were removed and counted 2–3 times per week.

Syrup consumption was measured by weighing the bottles of syrup 2–3 times per week. Vials were emptied and refilled with fresh syrup weekly. Consumption per bee was calculated as the observed consumption for a given cage divided by the number of “bee-days” for that time period, calculated as the product of the average bee density during a time period and the length of that time period in days. Consumption data in cages with fewer than an average of 10 bees were removed from the analyses due to high error relative to the mean. After five weeks, all remaining bees counted in each cage. The entire experiment was repeated the following August.

### 2.4. Statistical Analysis

The area of sealed brood per frame was measured from photographs using CombCount [21]. Continuous hive weight data per day (from midnight to midnight) were fit with piecewise regression using R version 3.6.1 (R Development Core Team, 2020), with 100 iterations per day sample, yielding estimates for 4 break points, 5 slope values and the adjusted r^2^. Three parameters were used: (1) Night slope (rate of hive weight change, usually due to moisture loss, from midnight until dawn), (2) dawn break point (start of daily hive active period), and (3) slope of the 1st segment after dawn (rate of morning hive weight change). The slope of the 1st segment after dawn was attributed to both forager departure and moisture loss (i.e., nectar drying, respiration). Weight loss due to moisture was estimated from the night segment and its effects were removed from the slope estimate, resulting in the estimate of the initial forager population. Days with rainfall >3 mm, dates of hive evaluations, days with slopes <−0.4 kg/min or >0.4 kg/min, and days with forager weight change >0 (indicating hive weight gain) were excluded. To focus on the active season, analyses of hive weight data were limited to the three months after the end of treatment.

Temperature data were transformed into daily average and within-day detrended data, calculated as the difference between the 24 h running average and the raw data. Sine curves were fit to 3-day subsamples of detrended data, and amplitudes from those curve fits were used as a response variable [22] in C++ (Qt Creator 4.1.0). Three-day samples were chosen to both ensure sufficient data (to reduce the impact of outliers) and maintain sensitivity to shorter-term changes.

Data from hive assessments, hives scales, and temperature and CO_2_ sensors were evaluated in repeated-measures MANOVAs with treatment group, sample day and their interaction as fixed effects, hive number as a random effect, and an autoregressive (ar(1) or arma(1,1)) variance model. For the forager population, daily hive weight change, and average daily temperature, estimated adult bee mass from the pre-treatment hive inspection date was used as a covariate to control for pre-existing differences. Proc Univariate was used with all response variables to inspect the data for normality. Data analysis was limited to those days on which valid data for at least one hive in each treatment group were included. Varroa mite fall was log-transformed and analyzed using ANOVA with pre-treatment mite fall as a covariate.

For statistical analysis of the cage study temperature, data were limited to the first six hours after the incubator temperature dropped to 15 °C, in order to focus on when cluster temperatures were highest. The temperature data were then analyzed with treatment, experiment, day and all two-way interactions as fixed effects and cage number as a random effect. Similarly, average syrup consumption per bee per day during the course of the experiment was analyzed with temperature, experiment, and their interaction as fixed effects. Bee survivorship in cages was analyzed using the mixed-model Cox regression package coxme in R, with the same model of fixed and random effects as that for syrup consumption.

## 3. Results

### 3.1. Flonicamid Concentrations

Flonicamid concentrations conformed to a large degree to the expected concentrations (Table 1). Honey sampled pre-treatment had no flonicimid but did have flonicamid at the approximate sugar syrup concentrations in November. By February, concentrations were higher, as expected, as the honey dehydrates over time.

### 3.2. Field Study

Examples of brood frame photographs are shown (Figure 1). Exposure to flonicamid at 50 and 250 ppb had no observable effect on any of the response variables measured here (Table 2, Figure 2). Sharp changes in hive weight, such as at the end of September (see Figure 2), were attributed to changes in group means due to the death of colonies and their removal from the study. Variables measured at discrete intervals, i.e., adult bee mass and surface area of sealed brood, were similar among treatment groups (Figure 3). While these variables were measured on 4 occasions post-treatment, the final measurement in February was excluded because of excessive mortality of the colonies post winter: Both the control and 50-ppb treatment groups lost 4 of 6 colonies while the 250-ppb treatment group lost 2 of 6 colonies. A large part of the problem were likely Varroa mite levels—average mite drop rose from 24 ± 3 mites per day in June to 150 ± 25 mites per day by September, in spite of treatment in June.

### 3.3. Cage Studies

Bee cluster thermoregulation and sugar consumption per bee per day were unaffected by treatment. Average syrup consumption per bee per day did differ a great deal between the two experiments (see Table 2). Bees consumed 29.1 ± 0.6 mg per day of syrup on average in the July cage study, and 19.7 ± 0.7 mg per day on average in the August cage study, and the experiments were significantly different (F = 111.83, *p* < 0.0001). This was attributed to the additional pollen patty provided to cages in the second cage experiment. Analyses were conducted separately for each experiment, and no treatment effects were observed (Table 3). Survivorship was also significantly different between the two studies (z = −7.9, *p* < 0.0001), so survivorship with respect to treatment was considered separately for each study (Table 4). Pairwise contrasts of the different treatment groups revealed a significant difference between the 50 ppb and 250 ppb treatment groups in bee survivorship in the second cage experiment, but it is difficult to interpret as the result was not observed in the first experiment. 

## 4. Discussion

Bee colonies, in general, are easy to access and exhibit particular colony-level behaviors, such as foraging activity, thermoregulation, and CO_2_ management, so they are rich environments for the use of sensors. Response variables are often not limited to, say, average daily values, but instead are modeled using different approaches to obtain further information on between-day and within-day effects. Continuous weight and temperature data, for example, have been modeled using piecewise regression and sine curves, respectively, to obtain additional information and parameters from those curve fits used as response variables. Groups of worker bees in cages, without a queen, can also exhibit aspects of colony behavior, such as thermoregulation, and cage studies have also been effective at showing effects of sublethal pesticide exposure.

In previous work, this system of colony monitoring has demonstrated significant effects of several pesticides on one or more response variables, even at very low (5 ppb) concentrations [15,16,17,18,19]. Effects have been observed with respect to continuous data on hive weight, temperature, and CO_2_ concentration. In this study, we used the same approach. Treatment concentrations were derived from observed contamination levels in hive products sampled from commercial apiaries. However, no effects were observed with respect to hive assessment data, or continuous hive data, or survivorship or thermoregulation in cage studies. These results were unexpected. We did not gather data on learning, mobility, disease resistance, or foraging behavior of individual bees, which have also been found to be affected by sublethal pesticide exposure [23,24,25], but it can be argued that the colony is the functional economic unit of beekeeping and therefore a logical subject for the study of sublethal effects.

The lack of a significant effect of sublethal exposure of flonicamid on colony growth or behavior, or worker longevity, should not be taken as evidence that flonicamid is completely harmless to honey bees, but rather this study found no reason for concern within the relevant parameters. Another factor to consider are the health concerns of other pollinators, including other bees, wasps, flies, and butterflies. The impacts of flonicamid on those pollinators may be different and need to be taken into account.

One factor in the field study which would have added to experimental variance was high Varroa mite levels, in spite of treatment in June and again in October. The high Varroa levels no doubt contributed to low colony survivorship: Two colonies in the low flonicamid treatment group had died by the end of October, and a total of eight colonies had died by the end of the experiment in mid-February. Cage studies were conducted twice, and those experiments were significantly different with respect to food consumption and survivorship. Part of the reason for those differences likely had to do with different amounts of pollen diet given in the two studies. However, significant differences in the outcome of replicate cage studies have been reported elsewhere (e.g., [26]).

In conclusion, exposure to field-realistic concentrations of flonicamid did not have any measurable effects on honey bee colony growth or behavior, nor did it affect longevity, food consumption, or thermoregulation of worker bee groups in cages studies. While it is important to note sublethal effects when they occur, we believe it is important to note when they are not detected, in order to support the use of compounds, when compounds need to be used, that have minimal impact on these pollinators.

## Figures and Tables

**Figure 1 insects-13-00845-f001:**
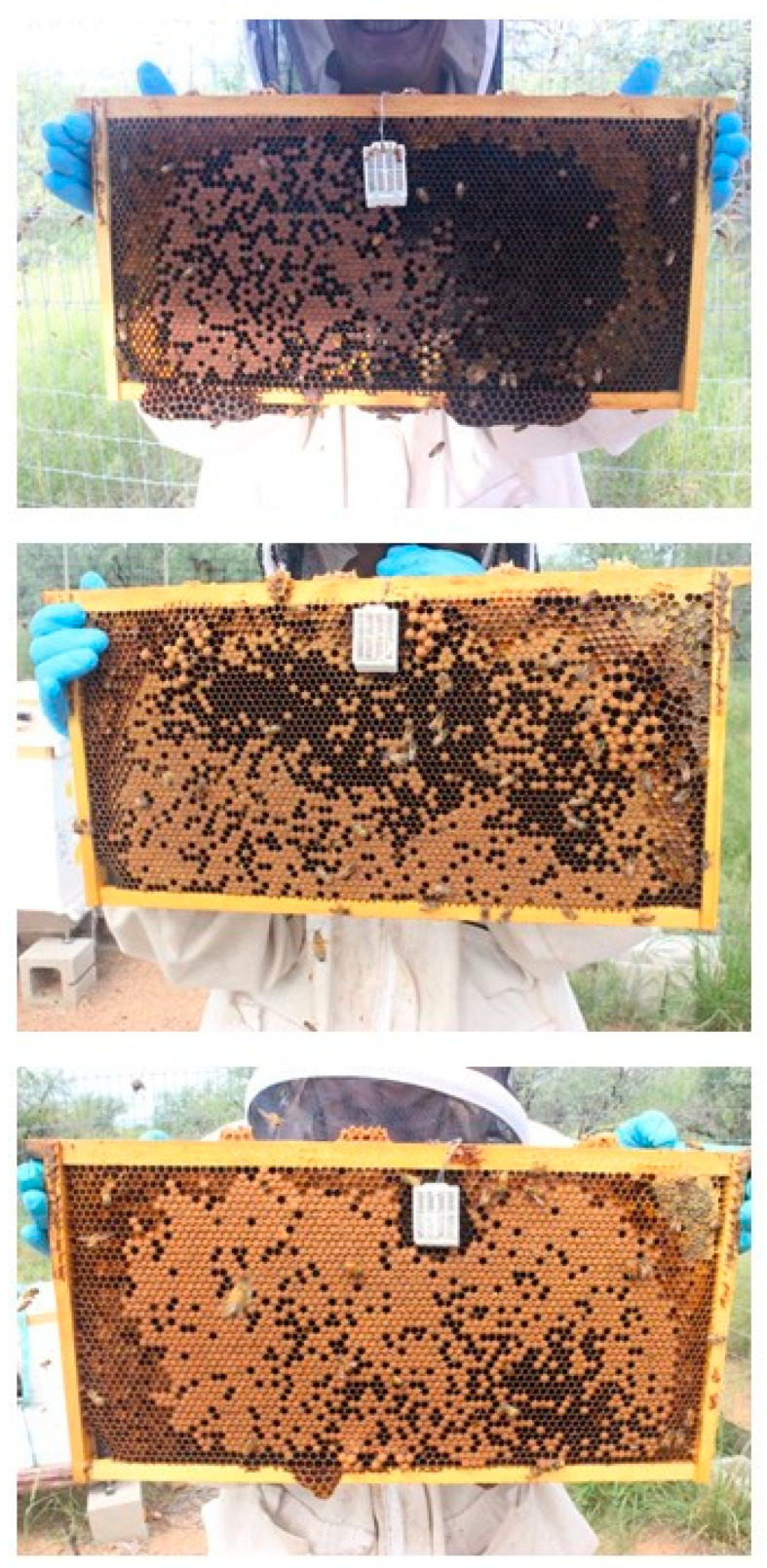
Examples of photographs of brood frames, including temperature sensor, taken on 6 September 2019, after the end of the treatment period. **Top** photo: Control treatment; **middle** photo: Low (50 ppb) exposure to flonicamid; **bottom** photo: High (250 ppb) exposure to flonicamid.

**Figure 2 insects-13-00845-f002:**
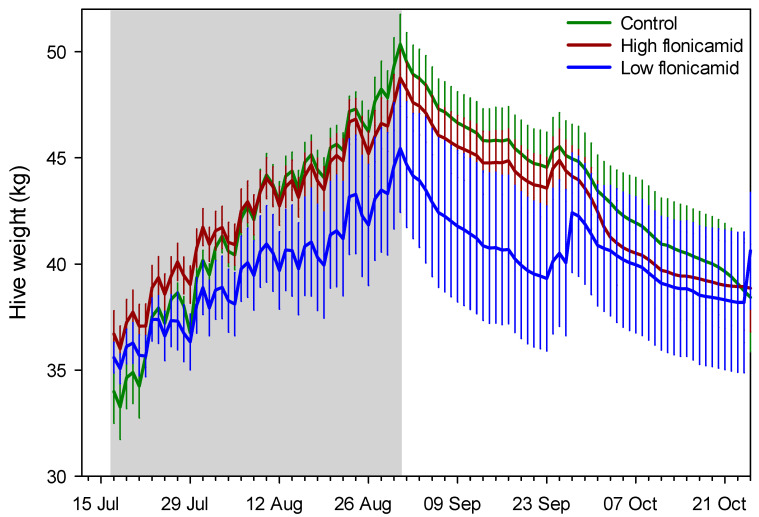
Daily mean (SE) hive weight for the three treatment groups, “High flonicamid” (250 ppb); “Low flonicamid” (50 ppb) and “Control” (0 ppb). The gray zone indicates the treatment application period. Shown are data until the end of October, which typically marks the end of the active foraging season in southern Arizona.

**Figure 3 insects-13-00845-f003:**
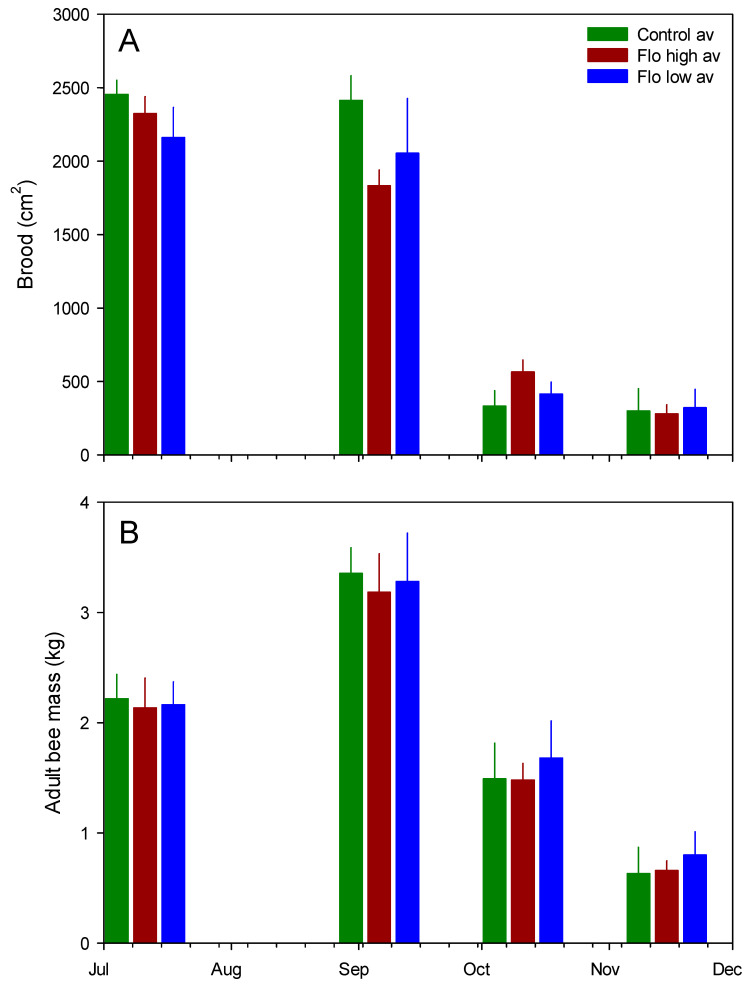
Mean (SE) colony measures across treatment groups. (**A**) Brood surface area; (**B**) adult bee mass.

**Table 1 insects-13-00845-t001:** Flonicamid concentrations in honey (sampled from bee hives) and in the sugar syrup treatment.

Matrix	Treatment Group	Pre Treatment	November 2019	February 2020
Honey	50-ppb flonicamid	0	58	112
	250-ppb flonicamid	0	272	360
	Control	0	Tr	Tr

Syrup	50-ppb flonicamid	67		
	250-ppb flonicamid	266		
	Control	0		

Tr = trace (<3 ppb).

**Table 2 insects-13-00845-t002:** Statistical results for flonicamid treatment.

Experiment	Response Variable	Factor	Num. d.f.	Den. d.f.	F	P
Field trial	Log adult mass	Treatment	2	13.53	0.10	0.908
	Log brood	Treatment	2	12.00	0.41	0.671
	Forager mass	Treatment	2	273.8	1.45	0.236
	Daily hive wt. change	Treatment	2	76.06	1.23	0.298
	Daily temp.	Treatment	2	13.08	0.46	0.641
	Log temp amplitude	Treatment	2	13.95	0.69	0.519
	Daily CO_2_	Treatment	2	22.15	0.99	0.389
	Log Varroa fall	Treatment	2	13.00	2.48	0.123

Cage trials	6 h cluster temp.	Treatment	2	41.84	1.29	0.285
		Experiment	1	41.84	2.10	0.155

“num. d.f.” means numerator degrees of freedom; “den. d.f” means denominator degrees of freedom; “F” means F statistic and “P” means probability of the observed F statistic.

**Table 3 insects-13-00845-t003:** Post hoc t test statistics and probability (*p*) values for consumption per bee per day for the two laboratory cage studies.

Contrast	1st Cage Study		2nd Cage Study	
	t	*p*	t	*p*
Control vs. 50-ppb flonicamid	−0.73	1.00	−0.03	1.00
Control vs. 250-ppb flonicamid	−1.62	0.36	0.20	1.00
50-ppb FLO vs. 250-ppb flonicamid	−0.89	1.00	0.23	1.00

**Table 4 insects-13-00845-t004:** Z scores and probability (*p*) values for the Cox regression survivorship analysis for two laboratory cage studies.

Contrast	1st Cage Study		2nd Cage Study	
	z	*p*	z	*p*
Control vs. 50-ppb flonicamid	−1.30	0.19	1.21	0.23
Control vs. 250-ppb flonicamid	−1.14	0.25	−1.09	0.28
50-ppb FLO vs. 250-ppb flonicamid	0.21	0.83	−2.19	0.028

## Data Availability

The datasets generated during the current study are available from the corresponding author on reasonable request.

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
