# Peer review of "Field and Cage Studies Show No Effects of Exposure to Flonicamid on Honey Bees at Field-Relevant Concentrations"

_insects, 2022, doi:10.3390/insects13090845_

Round 1

Reviewer 1 Report

Overall, I felt this paper was incredibly well-written, including a thorough introduction, clear and well-explained methods and results and a discussion which summarizes the data while still explaining what is left to be done.

The authors did a great job explaining all the variables which were used in the experiment. Although the paper found no difference between the control and treatment groups, I still believe this finding is incredibly important and will be of great interest to many readers.

I had a few minor comments/clarifications that I have noted on the attached document.

Author Response

Page: 1

I'm afraid I don't know why the text was highlighted. We did add "...in syrup..." to the part about 250 ppb flonicamid.

Page: 3

When was the pesticide treatment introduced and for how long?

Author: Please see lines 140-141

what are these treatments for?

Author: We now state "...for Varroa..."

Page: 4

why were these dates selected?

Author: We now state "Dates were chosen to obtain sufficient data without frequent colony disruption."

When was the pesticide treatment introduced and for how long?

Author: Please see lines 140-141

Page: 6

Why is there flonicamid in the syrup before treatment?

Author: This is an awkward concession to the table structure. What we meant was the the syrup samples were collected before feeding the bees, and those samples sent for residue analysis.

Page: 7

What is the grey indicating?

Author: Very good question - we had forgotten to put that in. We now state "The gray zone indicates the treatment application period."

Page: 8

This figure could be updated to be more visually pleasing. I do not have any specific suggestions, but currently it is difficult to get across the main point of the figure, which is there is no effect of the treatment.

Author: We did explore other ways to make the figure more agreeable, but it is true: really, there were no treatment effects so the only real thing to take away are the changes in adult and brood levels over time in southern Arizona.

Page: 9

was survivorship higher in the 50ppb or 250 ppb group?

Author: It was numerically higher in the 250 ppb group but the effect was not significant. We stated this in lines 233-35: "both the Control and 50-ppb treatment groups lost 4 of 6 colonies while the 250-ppb treatment group lost 2 of 6 colonies."

In previous studies, did they find that learning, behavior etc were affected when other hive assessment data was affected?

Author: Depends on the study, but in general studies that measure treatment effects on individual bees (e.g., foraging behavior, learning) seldom provide hive assessment data. However, readers are invited to check those references to see.

Page: 10

were these control or treatment hives?

Author: Please see lines 233-235

Reviewer 3 Report

Overall, the manuscript is good, but to increase the value of your work, please answer/solve/clarify the following:

For title: Lines 2-3

It seems more like a conclusion. Maybe it would be good to reformulate so that the reader is interested in reading the article... something like that: <Effect of exposure to flonicamid on bees at field relevant concentrations >

I recommend not to expose the conclusive result in the title.

For simple summary/abstract/key words:

Line 16: All pesticides? Or only those targeted, meaning neonicotinoids like flonicamides? You should mention something like...targeted pesticides...

You mention ...largely...what are you referring to? Please provide exact data, i.e. what are these?

Line 19:  Your studies are very important for farmers, as such you should highlight the results in relation to something that was not sufficiently studied before or, on the contrary, come up with the opposite. That's why I suggest you insert 1-2 sentences at the beginning of the Abstract to show what currently exists in the field and why this study is necessary

Line 34: The essential word is missing, i.e. honey bees

Maybe you should enter it in the set of keywords

Let's not forget that the journal you applied for is called Insects

And anyway, neonicotinoids are debatable as a large-scale use, in Europe, for example, several types of neonicotinoids are on the list to be taken out of use due to toxicity even in small quantities

So pay close attention to the importance you give them

And avoid the term...<we>.... esspecialy in the Abstract/Simple Summary

As a rule, the third person (in the past tense) is used, not the first person. Valid for the entire manuscript.

For example in Line 9: Instead of ...We had found a pesticide...you can use ...In our study it was found that....

For Introduction

Lines 47-48: You mentioned website addresses such as:https://pubchem.ncbi.nlm.nih.gov/com-47 pound/Flonicamid)

As a rule, website addresses are not mentioned in the text, but are integrated as a reference with the name of the article, the institution, then the website address and the date of access.

Line 55: same as the previous comment

Lines 59-60: As a suggestion when mentioning: Other neonicotinoid pesticides...

Please mention in parentheses what they are so that the reader does not have to look for the source

Line 61: I suggest adding 2-3 more sources and information because there are many in the field, to reinforce the need for your study. Come with results close to what you have obtained in this work.

Line 62: You should add a phrase that makes the connection between the results of previous studies and what you want to do. In fact, you should answer the question: Why did you want to do these studies? Did the previous ones not satisfy you? Want to check what others have achieved because you weren't quite sure it was?

For Materials and Methods

At: Syrup preparation

Lines 71-89: The method is very detailed. It is a very good thing. Does it belong to someone or does it belong to you? Should you mention if you used a standard method or your own? If it is usual, then name it and mention a source that can be found in the final references.Place the experiment in time. When did it happen (days? months?) recently or a while ago.

Line 89: I would suggest an explanation of the choice of this type of neonicotinoids, either integrated in the existing subchapters or in a separate subchapter. Why these and not others?. You mentioned in the Introduction that other neonicotinoids were used in research. Point out once again the reason for the choice to give strength to the methodology approached. And if it's the first time someone approaches you, then point this out.

Line 122: You talk about photographs and you focus a lot on this aspect. Then you should put some such images for credibility.

Because you are addressing a journal called Insects, you should treat the insects (bees) separately and make more references to the bee colonies used, characterize them more. Some pictures of the bee colonies would be welcome.

For Results

To Table 1/ Line 218: What does ppb mean? Explain the abbreviation so that those interested understand what you are talking about.

May be...parts per billion

To Figure 1: Remind in the legend or in the description of the Figure what the three groups are

Make the information in the tables clearer and easier to understand. Explain immediately below the tables each abbreviation in the table header or other abbreviations present.

To Discussion

Lines 255-293: The discussions are consistent without being redundant. However, comparisons with other studies are lacking. Please find some research with similar results and treat them according to the results of your work. Reinforce your results with clear arguments of what you brought in addition or what is similar.

Line 294: You should end with a final conclusion...In conclusion....and a recommendation. Please be objective and recommend others what to  others what to consider in the future.

For References

The references are few and others should be added (as I suggested in the Introduction and Discussions)

Author Response

The paper examined the effects of long-term exposure of honey bees to sublethal concentrations of floni-19 camid in a field study and two cage studies. However, there are several conceptual, methodological and procedural issues that need to be addressed as follows:

  1. The Summary, Abstract and Introduction sections are well written. However, in the introduction section, studies on critical local environmental conditions, such as temperature, season, relative humidity, wind, rainfall and CO2 levels, could be reviewed in context. They are missing.

Authors: We added text and references on the environment of the study.

  1. The Material and Methods section has several conceptual issues that need justification for the results to be more robust and convincing.

a. First, in line 93, “Each hive had at least 2 frames of sealed brood” needs to be made more explicit by specifying the exact number of frames of sealed brood. This way the field experiment will be replicable.

Authors: The number of frames containing sealed brood did vary among hives, which is normal (this was several months before treatment). We now specify the number of frames more exactly

b. Second, please explain why the hives were facing south – line 100.

  • Authors: We now explain that.

c. Third, temperature was recorded every 30 minutes (line 104) while other parameters were recorded every 5 minutes (line 98). It would have been interesting to monitor all the parameter at the same interval. An explanation needs to be provided.

  • Authors: We now provide the reason (data storage on temperature sensors was limited).

d. Fourth, please specify the type or source of the smoke – line 121.

  • Authors: We now specify the kind of smoke.

e. Line 130- please give a brief explanation of the choice of -80deg C.

  • Authors: That was actually incorrect. Samples were kept at -20C. We now state that and we state why (to preserve the samples for residue analysis).

f. Line 187 - Days with rainfall >3 mm considered: was it rainfall during the day or at night since night time rainfall may have no effect? A comment of the diurnal variability would be interesting.

  • Authors: Actually, rainfall at any point of the day interferes with the fit of the piecewise regression to the daily data. To see why we recommend the original paper on the subject (as cited).

g. Line 193 – An explanation 3-day subsamples used with supporting reference would be useful. 5-day averages (pentad rainfall) are common in literature, though.

Authors: We now explain why we use 3-day subsamples. We have tried samples of other sizes and found 3-day offered the best balance between robustness and sensitivity. Please see our paper on the subject (cited).

  1. The Results section presents interesting results.
  2. Discuss section is concise. However:, Lines 279 and 280 may suggest that the study is parochial. I recommend that it be re-worded.

Authors: We removed the phrase about the study being limited in time and resources (all studies are).